# Study on Characteristics of $^{125}$I Absorption and Accumulation in Eggplants

Chun-Lai Hong [1,*], Xin Lu [1], Huan-Xin Weng [2], Wei-Ping Wang [1], Feng-Xiang Zhu [1] and Yan-Lai Yao [1]

1 Institute of Environment, Resource, Soil & Fertilizer, Zhejiang Academy of Agricultural Sciences, Hangzhou 310021, China
2 Institute of Environmental and Biogeochemistry, Zhejiang University, Hangzhou 310027, China
* Correspondence: hongcl@zaas.ac.cn; Tel.: +86-0571-86409538

**Abstract:** Iodine fortification of plants is a means of improving the nutritional iodine status for humans. However, knowledge regarding iodine absorption and accumulation in plants remains limited. Hence, we used nutrient culture and isotope tracking methods, and the radioactivity of $^{125}$I was measured by using a multi-channel spectrometer to study the characteristics of $^{125}$I absorption and accumulation in an eggplant. The results showed that $^{125}$I was detected in the stems and leaves after 20 min of incubation in the iodine-containing nutrient solution, while it took 40 min to be detected in fruits, indicating a relatively slow migration of $^{125}$I from the roots to the fruits. The absorption and accumulation of $^{125}$I by various organs of the eggplant significantly differed, with $^{125}$I accumulation in the roots accounting for more than 80% of the whole plant (120 h), significantly higher than the above-ground parts. The absorption rate of all parts of the eggplant rapidly increased within a short period of $^{125}$I treatment, peaking at 12 h for the roots and at 3 h for transferring to the stems, leaves, and fruits. The results of this study indicate that the transferability of $^{125}$I from the roots to the over-ground parts of the eggplant is high, and eggplant fruits have the potential to continuously accumulate $^{125}$I. It is feasible to select eggplant as an iodine-rich crop for cultivation.

**Keywords:** $^{125}$I; eggplant; absorption; accumulation

## 1. Introduction

Iodine is an essential element of the synthetic thyroid, playing an extremely important role in the metabolism of human beings and animals [1]. The lack of iodine can cause goiter and other diseases, seriously influencing the intelligence development of children [2,3]. In 1983, Dr. Hetzel from Australia generally referred to these diseases caused by the lack of iodine as iodine deficiency disorders [4]. According to the previous research, except in Iceland, IDDs are common in countries all over the world, and about 2 billion people are under the threat of IDDs [5]. In 2003, it was reported that the ward population in China reached 400 million [6].

Beginning in 1994, the Chinese government recommended that all citizens should intake iodized salt, and the provision worked well [7–10]. However, there were many problems and deficiencies during practice. For example, inorganic iodine is easily volatile during transport and cooking, especially for iodine that has a bilateral threshold [11–15]. On the other hand, in certain areas of China such as the Xinjiang province, the local people still use local salt without iodine [16]. Under normal conditions, over 80% of the iodine in human bodies comes from plant-based foods. Furthermore, the natural intake of organic iodine from plant-based foods is better for maintaining the iodine balance in humans [17]. Therefore, other than adding iodine to salt, new methods of supplying iodine should also be discovered. On this condition, an agricultural biological enhancement method was employed to seek and cultivate an edible plant rich in iodine [18,19]. Previously, iodine was viewed as a radioactive contaminant in the environment. For example, in the repercussions

of the Chernobyl and Fukushima atomic control plant mishaps, huge sums of radioactive iodine were discharged into the air [20,21]. In fact, iodine shows a complex biogeochemical behavior in the environment; it has a high availability in soil, and the transfer of iodine from the soil to plants is known to be possible [22].

Except for marine plants, many terrestrial plants such as vegetables and rice also have a certain ability for accumulating iodine [23–25]. Up to now, however, the mechanism of plants absorbing and accumulating iodine is still unclear, and the dynamic absorption characteristics of plants for iodine in particular have been unreported. The present study used the nutrient culture and isotope tracer methods to study the dynamic absorption characteristics and distribution of iodine in eggplants.

## 2. Materials and Method

### 2.1. Materials for Experiment

The eggplant (Hangqie No. 1) was purchased from a vegetable seed shop in the Zhejiang Province. The seeds were soaked with 1% (M/V) potassium permanganate for 20 min for disinfection purposes; afterwards, the seeds were taken out and repeatedly rinsed with tap water. After that, the seeds were evenly spread on clean gauze and cultured at a constant temperature of 30 °C in an incubator. After 80% of the seeds germinated, they were transferred to a quartz sand bed for establishing seedlings, and were regularly supplied with 1/2 Hogland solution (Table 1). When two true leaves developed, healthy and uniform seedlings were selected and transplanted into 5 L plastic containers having fresh Hogland nutrient solution. Each pot had 3 seedlings. Each plant was fixed with sponges, and the roots were naturally suspended in the solution for a 24 h ventilation. The pH was maintained at 5.5 by using 0.1 mol/L of HCl or 0.1 mol/L of NaOH for adjustment. The nutrient solution was replaced every 5 days. When the eggplant was cultured to blossom and bear fruit, the whole plant was used as the experimental material for the next step.

**Table 1.** Composition of the Hogland balanced nutrient solution.

| Component | Concentration (mmol·L$^{-1}$) | Component | Concentration (μmol·L$^{-1}$) |
|---|---|---|---|
| $KNO_3$ | 6.00 | $H_3BO_3$ | 10.00 |
| $Ca(NO_3)_2$ | 3.50 | $MnSO_4 \cdot H_2O$ | 0.50 |
| $KH_2PO_4$ | 1.33 | $ZnSO_4 \cdot 7H_2O$ | 0.50 |
| $MgSO_4 \cdot 7H_2O$ | 0.50 | $CuSO_4 \cdot 5H_2O$ | 0.20 |
| NaCl | 0.48 | $(NH_4)_6Mo_7O_{24}$ | 0.01 |

The Na$^{125}$I solution was provided by the Isotope Research Institute of China Atom Science Research Institute, with a specific activity of $80 \times 37$ MBq/mL. The radiochemical purity was greater than 99.9%.

### 2.2. Experiment Method

The experiment was processed in a plastic rectangular box with a size of $50 \times 80 \times 20$ cm. We added 5 L of the Hogland nutrient solution into the basin, and 4 similar strains of eggplant, each with 2 young fruits, were transferred into the basins. After that, 35 μL of Na$^{125}$I solution was added. The specific activity of $^{125}$I in the nutrient solution was 537 Bq/mL. During the experiment, water was added to make the depth of the solution constant. The indoor temperature during the experiment ranged between 28 and 36 °C.

### 2.3. Sampling

After introducing Na$^{125}$I, we collected random samples at 5 min, 10 min, 20 min, 40 min, 60 min, 3 h, 5 h, 12 h, 24 h, 48 h, 96 h, and 120 h from the treatment groups. One plant was taken for each time interval. The roots were washed with deionized water and then dried with an air drier and absorbent paper. The root, stem, leaf, and fruit were

separated and cut into pieces. One gram of the plant sample was weighed for radioactivity determination. All experiments were repeated 3 times.

### 2.4. Determination

The determination of radioactivity was processed by a BH1224 multi-channel spectrometer (Beijing Nuclear Instrument Factory, China). The spectrometer was equipped with an upside-down scintillation probe φ70 mmNal, which was installed in a lead shield room. The sampling vessels had their own φ75 mm × 110 mm disposable plastic sample cups, which were placed on the upside-down scintillation probe. The probe's working voltage was 631 V, with a closed value of 0.28. The specific activity of $^{125}I$ is the concentration of iodine per unit mass of different parts of the eggplant. Total activity is the amount of $^{125}I$ accumulated in different parts of the eggplant [26].

### 2.5. Statistical Analysis

Statistical analysis was performed by using the SPSS 25 software. A one-way analysis of variance test was used to analyze whether the differences between the values of $^{125}I$ in different plant organs at different sampling times were significant ($p < 0.05$ indicates that it is significant).

## 3. Results

### 3.1. Dynamic Absorption of $^{125}I$ in Eggplant

Figure 1 shows the absorption and transportation of the $^{125}I$ in solution to the overground of the plant. Radioactivity was found in stems or leaves after 10 min of $^{125}I$ treatment, but it took 40 min to find any traces of radioactivity in the fruits. This indicates that whereas $^{125}I$ may be quickly delivered from a solution to the stems and leaves, it takes a significantly longer time to reach the fruits. Each section of the eggplant shows a rise in iodine content over time, showing that it has the capacity to accumulate $^{125}I$.

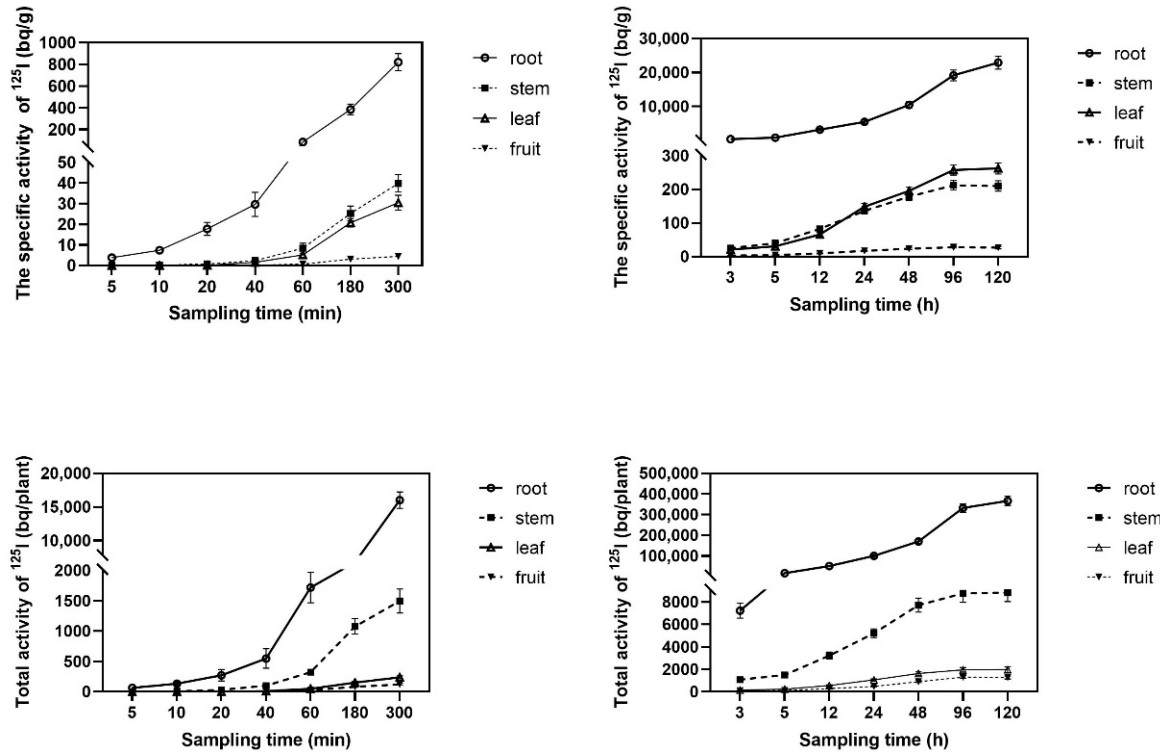

**Figure 1.** Dynamic absorption of $^{125}I$ in the eggplant. Specific activity = the total activity/per fresh weight. Values are given as the means ± SD (*n* = 3). The vertical lines show the data range.

Figure 1 further shows that there is a significant variation ($p < 0.05$) in the concentration of $^{125}I$ in various eggplant organs. The fruit had the lowest iodine concentration, and the roots had a substantially greater level of $^{125}I$ than the other organs. After 120 h of culture, the accumulation of $^{125}I$ concentration in the roots was 41.5, 186.4, and 286.6 times that of the $^{125}I$ concentration in the stem, leaf, and fruit, respectively.

### 3.2. Transport and Distribution of $^{125}I$ in Eggplant Plants

Figure 2 shows that the increased accumulation of $^{125}I$ by the eggplant with increased incubation time not only led to an increased accumulation of $^{125}I$, but also to a change in the distribution ratio of $^{125}I$ in each organ of the plant. The accumulation of $^{125}I$ in the roots as a percentage of the whole plant first gradually decreased, reaching a minimum (81.6%) at 1 h of treatment; after this time interval, the percentage gradually increased. In contrast, the proportion of $^{125}I$ distribution in the above-ground organs continuously increased during the first 1 h of $^{125}I$ treatment and then showed a decreasing trend.

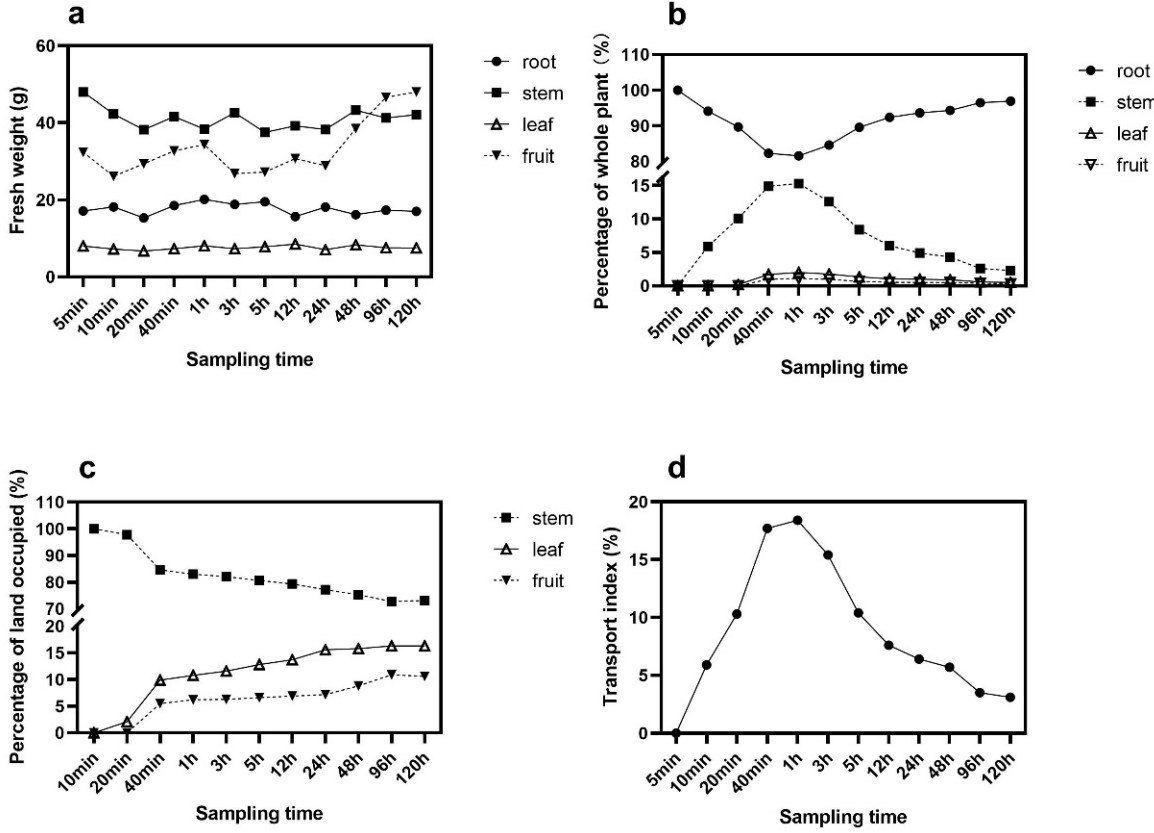

**Figure 2.** Transport and distribution of $^{125}I$ in the eggplant. (**a**) The variation of the fresh weight of different parts with time. (**b**) The proportion of $^{125}I$ in different parts of the whole plant with time. Percentage of whole plant = $^{125}I$ activity of a site/$^{125}I$ activity of the whole plant. (**c**) Proportion of $^{125}I$ in the above-ground sites with time. Percentage of land occupied = $^{125}I$ activity of a site on the ground/$^{125}I$ activity of the total site on the ground. (**d**) The variation of the transport index with time. Transport index = (total $^{125}I$ activity of above-ground parts/total $^{125}I$ activity of whole plant) × 100.

As the time interval continued, the amount of $^{125}I$ remaining in the roots increased and gradually transferred to the organs of the overground part (Figure 2). The transport index increased with time. At the 1 h interval for $^{125}I$ treatment, the transport index of $^{125}I$ from the root to the overground part reached 18.4%, and it gradually decreased as the processing time was prolonged. The proportion of the total transport amount of the overground part of $^{125}I$ from the root to the fruit showed an upward trend throughout

the whole processing period, indicating that the transport activity of $^{125}$I to the fruit of the eggplant was relatively high. With the prolongation of time, the edible parts (eggplant fruit) maintained the potential for accumulating $^{125}$I.

### 3.3. The Absorption Rate of $^{125}$I in Eggplant

Figure 3 shows that the absorption rate of $^{125}$I in various parts of the eggplant plant increased significantly within a short time ($p < 0.05$), especially after 40 min of treatment. The absorption rate of $^{125}$I in various organs of the overground part reached a peak at 3 h, and the root reached the highest value at 12 h of absorption. After these times, the absorption rate of $^{125}$I by various organs of the eggplant gradually declined. The absorption rate of the leaves and fruits decreased significantly less than that of the roots and stems. Additionally, the absorption rate of the roots was significantly higher than that of the above-ground parts, and was followed by the stems, leaves and fruits, indicating that the magnitude of $^{125}$I uptake in the whole eggplant depends on the absorption capacity of the eggplant's roots. This phenomenon may be mainly caused by the high concentration of $^{125}$I in the extra-root culture solution, which facilitates the passive absorption of exogenous $^{125}$I by the roots of the eggplant.

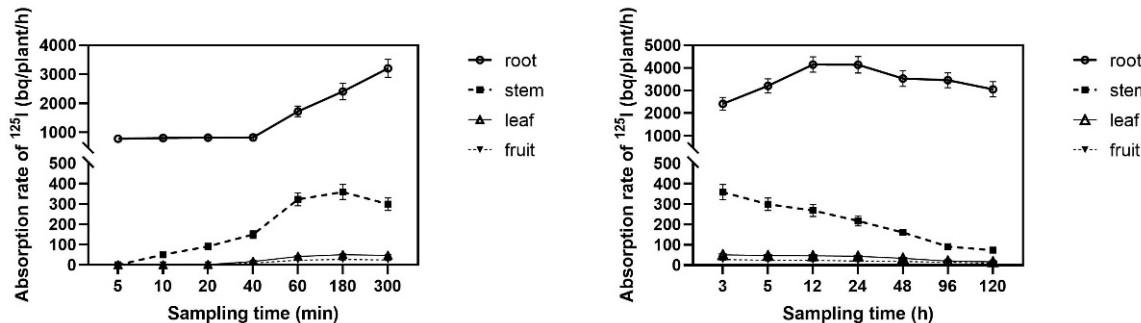

**Figure 3.** Absorption rate of $^{125}$I in eggplants. Values are given as the means $\pm$ SD ($n = 3$). The vertical lines show the data range.

### 4. Discussion

As the inadequate levels of trace elements in the environment are the primary cause of trace element deficiency in both humans and animals [27], soil scientists and nutritionists have long advocated the use of soil fertilizer management techniques; this has been supported to improve the essential trace element content in crops for the sake of increasing the population's intake of trace elements in areas where trace elements are deficient [28]. It has been proven in practice that this is an economical, safe, and scientific method for trace element supplementation [29–31]. Vegetables are indispensable food in daily life, as well as an indispensable and important source of essential nutrients and beneficial trace elements in the human body. The result of the dynamic absorption and accumulation of $^{125}$I by the eggplant indicated that it is possible to replenish iodine amounts in the human body through the ingestion of plants that are rich in iodine uptake. Studies on the mechanism of iodine uptake and accumulation by plants have found that iodine can enter plant root cells through active or passive means by using specific carriers or channels; after entering the plant interior, it is mainly transported to stems, leaves and fruits through the xylem pathway [32,33]. Iodine is transported via transporters and channels, including chloride ($Cl^-$) channels, $Na^+$:K+/$Cl^-$ cotransporters, $H^+$/$Cl^-$ cotransporters or antiporters, and $Cl^-$ transporters that are driven by ATP-dependent proton pumps, which may also be involved in iodine transport due to the similarity of the chloride and iodine ions [34,35]. It has also been shown that iodine can be transported through the siliques, but this transport route is less efficient because of the severely restricted mobility of the siliques [36]. However, the current research mainly focused on the pollution of radioactive iodine to the environment and its transfer into the soil–plant system. The oceans serve as a reservoir for the global

iodine cycle, and the most crucial component of the global cycle is the volatilization of iodine and its transit via the atmosphere into the terrestrial ecosystem. It is generally accepted that the main source of iodine in the soil is the atmospheric transport, with rainfall being the most important form of transfer [37]. Plants can take up iodine from the soil through their roots. In contrast, iodine in the soil is to some degree bounded, and this constrained iodine is taken up by the root system, which is closely related to their transfer factors [38]. Plants can also absorb iodine from the atmosphere through the stomata and epidermis of the leaves [39]. There are few studies on the absorption and accumulation of iodine by crops [40]. It is therefore necessary to further strengthen the systematic research in this field, which is beneficial in solving uptake-related problems.

## 5. Conclusions

In summary, the eggplant is able to absorb additional iodine through the roots and transport it to all parts of the plant above the ground. The fruit parts of the eggplant had a certain ability to continuously absorb and accumulate iodine. Although the rate of iodine transport from the roots to the fruits was slow, the accumulation of iodine in fruits could be increased by appropriately extending the incubation time. This further proves that it is feasible to select eggplants and other fruit vegetables as subjects for the cultivation of iodine-rich crops. In the long-term, this method can increase the level of iodine in the food chain of iodine-deficient areas.

**Author Contributions:** Conceptualization, H.-X.W.; data curation, X.L.; formal analysis, Y.-L.Y.; funding acquisition, C.-L.H.; investigation, Y.-L.Y.; methodology, C.-L.H.; supervision, W.-P.W.; validation, W.-P.W. and F.-X.Z.; visualization, F.-X.Z.; writing—original draft, C.-L.H.; writing—review and editing, X.L. All authors have read and agreed to the published version of the manuscript.

**Funding:** This work was supported by grants from the National Natural Science Foundation of China (40873058).

**Institutional Review Board Statement:** Not applicable.

**Informed Consent Statement:** Not applicable.

**Data Availability Statement:** The corresponding author will make the datasets that were generated to support the current study's findings available upon reasonable request.

**Conflicts of Interest:** The authors declare no competing interest.

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
