# Peer review of "Study on Characteristics of 125I Absorption and Accumulation in Eggplants"

_sustainability, doi:10.3390/su141912389_

Round 1
Reviewer 1 Report
This article by Hong et al. entitled “Study on characteristics of I125 absorption and accumulation by eggplant”. I have gone through this manuscript. In this manuscript, the authors have shown the characteristics of I125 absorption and accumulation by eggplant. This is a good work but the authors failed to explain this issue properly. There are many limitations such as abstract, results and discussion are poorly written; Materials and methods lack of scientific information detail. This part is totally short which loses reproducibility. The main limitation of this research that the authors didn’t mention the mechanism and pathway of I125 absorption and accumulation in the different parts of eggplant. In the discussion section, the authors should discuss the mechanism how the I125 absorp and accumulate in the different parts of egg plants. Moreover, the reportage needs to be look into again perhaps by Native English Editorial Service to make the rendering more meaningful. I wish you good luck.
Reviewer 2 Report
1. Please rewrite the abstract.
2. Please digest at least 10 recent references in the introduction about the importance Iodine.
3. please separate between discussiona and conclusions.
4. Please write small paragrph about your contribution to knowledge.
5.Please rewrite the Paragraph from line 143 to 147 ( The lack....etc).
6. pleaae rewrite the paragraph from line 110 to 112.
Reviewer 3 Report
Labeled atoms studies are very important in researches that aim to establish the quality of vegetable food as they are a very clear and comprehensive method in highlighting the way nutrition elements transfer from soil/nutritive solution to different organs of plants.
The Abstract very well synthesizes the research.
Very clear and well substantiated Introduction.
Could you elaborate on ”water planting”? I suppose it has to do with hydroponics, since you used the expression ”solution cultivation” in the Abstract, but I never encountered the term.
Very clear, right to the point, presentation of the materials, method, sampling, and analytical procedure. Statistical computing should also be added. Since samples were analyzed in three replicates I think an ANOVA statistical analyze would fairly complete the research by adding the statistical significance of the differences between the values of 125I in different plant organs at different sampling times. It also makes the results morecredible.
Figures 1 are clear and comprehensive in describing iodine intake of plants' organs. Explain what ”specific activity” and ”total activity” mean and how are they differentiated through measurement.
In Figure 2 each ”y” value (fresh weight, ... transport index) should be explained and its measurement/calculation presented.
Define ”absorption rate” and ”migration rate” for Figures 3.
Good and pertinent discussions, consistent with the findings and backed by literature.
Many and very up-to-date references.
Round 2
Reviewer 1 Report
This article by Hong et al. entitled “Study on characteristics of I125 absorption and accumulation by eggplant”. I have gone through this revised version manuscript and this is much better than the previous one. The authors addressed all issues of this manuscript properly. Therefore, I would like to appreciate their hard work. Now the research work may be accepted for publication.
Minor revision:
Line no 162: Discussion headline should be separated from figure legend.
